# Evolution of Inflammatory and Oxidative Stress Markers in Romanian Obese Male Patients with Type 2 Diabetes Mellitus after Laparoscopic Sleeve Gastrectomy: One Year Follow-Up

**DOI:** 10.3390/metabo10080308

**Published:** 2020-07-28

**Authors:** Ariana Picu, Laura Petcu, Diana Simona Ştefan, Grațiela Grădișteanu Pîrcălăbioru, Manuela Mitu, Daiana Bajko, Daniela Lixandru, Cristian Guja, Octavian Savu, Anca Pantea Stoian, Alina Constantin, Bogdan Smeu, Cătălin Copăescu, Mariana Carmen Chifiriuc, Elena Ionica, Constantin Ionescu-Tîrgovişte

**Affiliations:** 1NIDNMD “Prof. N.C. Paulescu”, 2nd district, 020042 Bucharest, Romania; arianapicu@gmail.com (A.P.); simona_ds2002@yahoo.com (D.S.Ş.); mani_mitu2002@yahoo.co.uk (M.M.); daianabajko00@gmail.com (D.B.); cristian.guja@b.astral.ro (C.G.); savu.octavian@gmail.com (O.S.); cit.paulescu@gmail.com (C.I.-T.); 2Faculty of Biology, University of Bucharest, 5th district, 050095 Bucharest, Romania; elena.ionica@g.unibuc.ro; 3Research Institute of the University of Bucharest, ICUB, 5th district, 050107 Bucharest, Romania; 4Department of Biochemistry, “Carol Davila” University of Medicine and Pharmacy, 5th district, 050474 Bucharest, Romania; danielalixandru@gmail.com; 5Department of Diabetes, Nutrition and Metabolic Diseases, “Carol Davila” University of Medicine and Pharmacy, 5th district, 050474 Bucharest, Romania; ancastoian@yahoo.com; 6Department of Doctoral School, “Carol Davila” University of Medicine and Pharmacy, 5th district, 050474 Bucharest, Romania; 7“Nicolae Simionescu” Institute of Cellular Biology and Pathology, 5th district, 050107 Bucharest, Romania; alina.constantin@icbp.ro; 8Metabolic and Bariatric Surgery Center of Excellence, Ponderas Academic Hospital, 1st district, 014142 Bucharest, Romania; bogdan.smeu@ponderas-ah.ro (B.S.); catalin.copaescu@ponderas-ah.ro (C.C.); 9Romanian Academy of Medical Sciences, 3rd district, 030171 Bucharest, Romania

**Keywords:** T2DM, laparoscopic sleeve gastrectomy, diet, conventional antidiabetic oral therapy, obesity, oxidative stress, incretin hormones, proinflammatory state

## Abstract

Geography is one of the key drivers of the significant variation in the etiopathogenic profile and prevalence of type 2 diabetes mellitus (T2DM) and obesity, therefore geographically based data are fundamental for implementing the appropriate interventions. Presently, the selection criteria of T2DM and obesity patients for laparoscopic sleeve gastrectomy (LSG) have not reached a worldwide consensus—highlighting the need for sharing experts’ guidance in the preoperative evaluation, choice of the interventional procedure, perioperative management and patient long-term care. The aim of the current study was to evaluate the impact of LSG on T2DM (T2DM) remission in Romanian obese male patients, based on a multiparametric, prospective investigation. We have conducted a randomized controlled study on 41 obese male participants with the body mass index (BMI) ≥ 30 kg/m^2^, aged 30–65 years, which were randomly divided in two study groups: one receiving conventional treatment and the second undergoing LSG. The clinical and anthropometrical parameters, resting metabolic rate, general biochemical status, adipocytes profile, gastrointestinal hormones levels, proinflammatory, oxidant and antioxidant profiles were determined at three time points: V1 (baseline), V2 (after six months) and V3 (after 12 months). Glycated hemoglobin (HbA1c), blood glucose levels, BMI, weight, visceral fat level, HDL-cholesterol, incretin hormones, proinflammatory and the oxidative stress status were significantly improved in the LSG versus conventional treatment group. This is the first study reporting on the evaluation of metabolic surgery impact on Romanian obese male patients with T2DM. Our results confirm that LSG could contribute to T2DM remission in patients with diabesity, but this beneficial effect seems to be critically influenced by the duration of T2DM rather than by the obesity status. Our results show that, in addition to the parameters included in the prediction algorithm, the proinsulin levels, proinsulin/insulin ratio and the visceral fat percentage could bring added value to the assessment of metabolic status.

## 1. Introduction

Presently, we are facing a worldwide epidemic of obesity and T2DM—often linked together, as suggested by the currently used term of “diabesity” [1]. The most recent global predictions of the International T2DM Federation (IDF) suggest that there are 285 million people with T2DM currently worldwide and could reach 438 million by 2030, [2], with a further half billion at high risk, T2DM being therefore one of the greatest public health threats of the 21st century. Premature mortality and morbidity in T2DM mainly result from microvascular and macrovascular complications, but also from other dysfunctions involving lipid metabolism, oxidative stress and inflammation. On the other hand, studies have shown that untreated obesity can shortly lead to T2DM. The excessive adipose tissue is associated with a chronic proinflammatory condition, contributing to the occurrence of insulin resistance, a fundamental pathogenic mechanism involved in the development of T2DM.

Current data show that the number of “diabesity” cases is alarmingly increasing in Romania, with nearly 2 out of 10 young people being overweight [3,4] and 20% of the entire population with obesity developing T2DM. According to the PREDATORR (Prevalence of T2DM and preT2DM in the adult Romanian population) study conducted in 2016, in about 48% of cases, T2DM is associated with morbid obesity [4].

The bariatric–metabolic surgery procedures, including the increasingly popular laparoscopic sleeve gastrectomy (LSG), are presently considered an efficient tool for T2DM control and remission in obese patients [5,6,7,8,9,10,11].

The aim of the current study was to evaluate the evolution of inflammatory and oxidative stress markers in male patients with T2DM and obesity underlying LSG, by performing a multiparametric, prospective investigation of clinical and anthropometrical parameters, resting metabolic rate, general biochemical status, adipocytes profile, gastrointestinal hormones levels, proinflammatory, oxidant and antioxidant profile, including respiratory burst determination in peripheral blood mononuclear cells, all determined prior to, as well as 6 months and 12 months after the LSG intervention.

## 2. Results

This is the first prospective randomized study performed on Romanian patients aiming to evaluate the degree of T2DM remission in obese individuals, after the bariatric–metabolic surgery in comparison with conventional treatment.

The patients included in this study were relatively young, with a mean age for conventional treatment group of 48.7 ± 6.8 years and of 46 ± 5.9 for the LSG group, with morbid obesity and poor metabolic control for both groups, despite the relatively short duration of T2DM, i.e., 6.3 ± 4.5 years for the conventional group and 5.4 ± 2.9 for the LSG one. No statistically significant differences in the age and duration of the T2DM between the studied groups were found. Table 1, Table 2 and Table 3 from Appendix A show the results of the clinical, anthropometrical and metabolic status evaluation of the patients from the two study groups prior to LSG (V1) and 6 months (V2) and 12 months (V3) after bariatric surgery intervention. The results of the multiparametric investigation for both studied groups obtained at V1 were used as reference and compared to those obtained at V2 and V3, respectively.

There were no statistically significant differences between the parameters determined at baseline for the two studied groups, which demonstrates the validity of the randomization algorithm used in our study (Appendix A).

Six months after LSG, patients from the surgical group showed statistically significant (*p* < 0.05) and clinically relevant improvements for 11 out of 19 tested clinical and biologic parameters versus only two (BMI, waist circumference and systolic blood pressure) in the conventional treatment group, confirming the therapeutic efficacy of this surgical procedure (Appendix A).

The evaluation after 12 months indicated that patients from the surgical group have still shown statistically significant improvements (*p* < 0.05) for 17 out of 24 tested clinical and biologic parameters, including a remarkable improvement of BMI, waist circumference, metabolic control and lipid profile versus only 7 in the conventional treatment group (Appendix A). The conventional treatment group showed a statistically significant decrease only for the predicted baseline metabolism (RMR), while patients in the surgical group presented significant changes for both predicted and measured RMR. In T2DM with obesity, RMR is higher than in obesity alone, due to increased glucose oxidation, decreased glucose storage and increased sympathetic nervous system activity [12]. The patients from the conventional treatment group showed a paradoxical weight gain, probably due to the loss of motivation to respect long-term treatment, diet and physical exercise recommendations (Appendix A).

Concerning the hormonal levels (Table 1), insulin, proinsulin and proinsulin/insulin ratio (an indirect indicator of the degree of beta–cell dysfunction), pointing an improvement in glycemic control exhibited a decreasing trend along the 12 months of evaluation in both groups, but all these parameters improved more significantly in the surgical patients. The same dynamic was observed for leptin, the decreased levels being correlated with the decrease in BMI.

Patients in the surgical group exhibited a statistically significant decrease along the entire monitorization period in HOMA-IR, C peptide and circulating levels of ghrelin, the orexigenic hormone, correlated with the weight loss. In addition, the patients in the surgical group experienced a significant increase in adiponectin (probably contributing to the improvement of insulin resistance) and a decrease in GLP-1.

The proinflammatory and oxidative stress markers were evaluated post-surgery, at V2 and V3. Concerning the proinflammatory status, although IL-6 and TNFα have shown increased values at 12 months (V3) than V2 in both groups, however the significant decrease of the hsCRP values in surgical patients (*p* = 0.0006) indicates an improvement of the subclinical inflammation following LSG (Table 2). The level of homocysteine remained unchanged in both the surgical and conventional groups, indicating that this biomarker is not useful in assessing candidates for bariatric surgery (Table 2).

The analysis of the oxidative stress markers including respiratory burst and glutathione levels pointed unfavorable outcomes for the surgical group, as compared with the dietary and intensive medical therapy group (Table 3).

The levels of antioxidants markers in erythrocytes obtained from the patients included in the study are presented in Table 4.

## 3. Discussion

T2DM is often associated with obesity, involving the contribution of common etiopathogenic conditions, such as the excess and distribution of white adipose tissue, the proinflammatory state and oxidative/antioxidative imbalance, which are combining, giving rise to particular clinical conditions [13,14,15]. LSG has been proposed as an optimal surgical intervention for patients with mild, non-morbid obesity and T2DM [16,17,18,19,20]. The risk of major postoperative complications after LSG is 5%–10%, which is lower than the risk associated with gastric bypass or malabsorptive procedures such as duodenal switch. Complications that can occur after LSG include a leak from the sleeve resulting in an infection or abscess, deep venous thrombosis or pulmonary embolism, narrowing of the sleeve (stricture) requiring endoscopic dilation and bleeding. Major complications requiring a new surgical intervention, are uncommon after sleeve gastrectomy and occur in less than 5% of patients. The mechanism for T2DM remission after LSG is not fully elucidated, involving not only a quick improvement in glucose homeostasis, but also a reduction of other co-morbidities like dyslipidemia, cardiovascular risk factors, hypertension and obstructive sleep apnea syndrome (OSAS) [20,21,22]. The results reported in the literature are confirmed by the present study, showing a statistically significant decrease in glycated hemoglobin (HbA1c), blood glucose, BMI, weight, visceral fat level, HDL-cholesterol and incretin hormones levels in LSG patients compared with the conventional treatment group [23]. As expected, insulin resistance (evaluated by HOMA-IR) was not significantly improved in the control group (from 8.12 to 6.32, *p* = 0.168), in comparison with the surgical group (from 8.44 to 1.19, *p* < 0.001). Results are comparable to those reported by Schmatz R et al. with mean HOMA-IR decreasing from 6.08 to 1.28 following bariatric surgery in a group of 20 obese T2DM subjects [24].

Obesity associated with T2DM enhances the proinflammatory status due to some specific features of the adipose tissue, like hypertrophy, hyperplasia, a peculiar fat distribution, predominantly abdominal (perivisceral), an increased secretion of adipokines and infiltration of with monocytes–macrophages inducing overloading of the endoplasmic reticulum (ER), which leads to ER dysfunction or ER stress, accumulation of incorrectly folded proteins, adipocyte apoptosis amplifying the inflammatory cascade [25]. An important weight loss can attenuate the proinflammatory response of abnormal, dysfunctional adipocytes regressing to their natural state of normal functioning adipocytes [26]. A very interesting finding is related to the increased levels of proinflammatory cytokines IL-6 and TNF- α in both studied groups at V3, than V2, despite the continuing loss of adipose mass recorded in both groups. On the other hand, the levels of C reactive protein (hsCRP), an acute phase molecule involved in the early stages of the inflammatory process has shown a continuously decreasing trend from V2 to V3 in the LSG group (*p* < 0.0006447). One limitation of the present study is the short period of post-surgery evaluation. Tracking patients in follow-up for more than one year would have brought more information about the trend of decrease/stabilization of the cytokines levels.

The statistically significant decrease of fat mass and of visceral fat level leads to a significant decrease in the secretion of leptin (*p* < 0.001) and an increase in adiponectin levels, resulting in insulin sensitivity, in the surgical group. The decrease of leptin levels in LSG patients was previously reported [27,28] and is explained by the decrease of fat tissue mass in parallel with weight loss. The levels of adiponectin were correlated negatively with the waist circumference (r = −0.49, *p* < 0.001), diastolic BP (r = −0.30, *p* < 0.05), levels of uric acid (r = −0.33, *p* < 0.001), glucose (r = −0.30, *p* < 0.05). A less pronounced, but also significant (*p* = 0.002) decrease of leptin was recorded in the conventional group.

The cardiovascular risk of T2DM patients is directly influenced by the lipid profile (TC, TG, HDL and LDL). In our study, we observed an increase in HDL cholesterol and a decrease in serum triglycerides in the LSG group, which can be considered slightly positive prognostic factors, although the total cholesterol and LDL cholesterol levels remained unchanged. Despite considerable and growing progress in understanding the mechanism responsible for improvement of dyslipidemia and insulin resistance after LSG, this remains largely unknown. Increased insulin sensitivity is usually associated with total weight loss and fat tissue reduction. However, the interpretation of lipid spectrum changes is not easy. An improvement of dyslipidemia and obesity-related comorbidities may be less dependent on weight loss, being rather due to hormonal changes after LSG. Another possible explanation could be that LSG reduces the gastric volume and consequently, the production of gastric lipase and cholecystokinin, that can lead to a drop in the triacylglycerols hydrolysis and of free fatty acids absorption. Overall, the percentage of excess weight loss (EWL) in the LSG group was 78.98% compared with only 9.46% in the control group.

The reduced caloric intake immediately after surgery is causing the decrease of adipose tissue mass—and also changes in the incretin hormonal status and in intestinal absorption—contributing to the remission of T2DM associated with obesity [20,21,22].

Among incretins, ghrelin and the glucose dependent insulinotropic polypeptide GLP-1 are the most likely candidates for increasing insulin sensitivity after this type of surgery, even before the occurrence of substantial weight loss. GLP-1 is secreted by endocrine L-cells in the mucosa of the ileum and colon and regulates glucose homeostasis in T2DM patients, by its concurrent insulinotropic and glucagonostatic actions [29]. In addition, ghrelin levels are reduced due to the resection of gastric fundus cells [30]. Decreasing the gastric volume has a direct consequence not only on lowering the level of ghrelin, but also in decreasing appetite and food intake [19].

Ghrelin is a growth hormone-releasing peptide, produced by the stomach and duodenum [31]. It is the only known orexigenic gut hormone. Additional evidence suggests that ghrelin may also contribute to long-term body weight regulation, being a potential therapeutic candidate for fighting obesity. Ghrelin contributes to preprandial hunger and meal initiation [32,33,34,35,36,37,38,39]. In our study, there were no statistically significant correlations between ghrelin and the other studied parameters.

Mainly seen as an indicator for impaired β-cell function, proinsulin can be detected at low concentrations in the blood of healthy persons, but is found at higher concentrations in the blood of insulin-resistant subjects [36] and patients with T2DM [33,34,35]. As expected, patients in the surgical group harbored a significant decrease of intact proinsulin (from 5.86 to 0.70 pmol/L, *p* < 0.001), indicating improvements of beta cell dysfunction or a beta cell “rest state” after metabolic surgery [40]. However, a significant decrease of intact proinsulin in the control group (from 6.42 to 3.06 pmol/L, *p* = 0.0011) despite the modest weight loss and metabolic improvement in these patients, is suggesting the positive impact of lifestyle changes on the beta cells function. The efficiency of β-cells to convert proinsulin to insulin depends on endoplasmic reticulum (ER) packaging or folding capacity, the available space for protein folding, the clearance rate of misfolded proteins to avoid accumulation of toxic debris and last, but not least, the RE ability to efficiently carry the folded proinsulin to the next secretory compartment (Golgi apparatus) [41]. Therefore, high levels of proinsulin indicate an advanced stage of pancreatic beta cells depletion and represent a highly specific marker for insulin resistance that can be used to determine the therapeutic decision in T2DM. Thus, the evaluation of beta cell secretion should include, besides insulin and C peptide serum levels, proinsulin, that provides valuable information on the location of the insulin-secreting defect and the progressive or regressive course of the pathogenic process in T2DM. Similar changes were found for the proinsulin–insulin ratio, also a valuable indicator of beta cell dysfunction, showing the relationship between the secretory demand and the secretory response of the pancreas [26,40,42,43]. The proinsulin to insulin ratio decreased in LSG subjects (*p* < 0.001), demonstrating that the secretory burden of the pancreas decreases, while the secretory response improves.

In T2DM, chronic hyperglycemia enhances oxidative stress involved in the occurrence of complications, consecutive to oxidative lesions appearing in different organs and systems, the result being the increase in free radical production (increasing the oxidant status and the respiratory burst) or a decrease in antioxidant capacity, mediated by superoxide dismutase (SOD), catalase (CAT), glutathione peroxidase (GPx), glutathione (GSH) and PON1. There are no longitudinal data about the redox homeostasis of patients undergoing bariatric treatment. Starting from the hypothesis that the huge weight losses after LSG could have an impact on the blood antioxidant systems/oxidative stress, we have evaluated these processes at 6 and 12 months. The analysis of oxidative stress markers, including RB, 8-OH-2dG and levels of antioxidant enzymes, after six months has also shown favorable results in the surgical groups, as compared with the group receiving intensive medical therapy alone.

While the levels of antioxidant enzymes GPx and CAT increased after six months in both groups, activities of SOD and PON1 (PON1phe and PON1dhc) were similar in the two groups, which suggests the persistence of a redox imbalance in obese individuals, even if body weight has been significantly reduced, as is the case in the LSG group.

Of the investigated parameters, PON1dhc activity correlated positively with concentrations of HDL-C and adiponectin (*p* < 0.05) and negatively with BMI, waist circumference, SBP, levels of HbA1C, insulin and HOMA-IR (*p* < 0.05).

The positive correlation between adiponectin and PON1 dhc remained significant even after adjustments for age, gender, BMI, blood pressure, HOMA-IR, HDL-C and LDL-C. These correlations highlight the role of PON1 in removing harmful oxidized lipids that are the major cause of inflammation in diabesity.

Regarding the glutathione (the most important intracellular antioxidant), its level was also significantly reduced in LSG patients at six months after the bariatric treatment in parallel with an increase in respiratory burst. This indicates a higher depletion of glutathione reserves in LSG patients, probably due to the overproduction of ROS and the enhanced cellular oxidative damage. Moreover, despite the proven effectiveness of bariatric surgery, it is unclear whether it improves the redox homeostasis of diabesity cases.

Debates still exist on which factors play an important role in predicting the outcome of bariatric–metabolic surgery on T2DM associated with obesity remission. The ABCD score includes age, BMI, C peptide and duration of the disease, while DiaRem includes age, HbA1c, antidiabetic drugs and insulin [44].

Our results indicate that other parameters should also be considered for predicting the T2DM remission probability after metabolic surgery. Lower values of HDLc are correlated with a lower probability of T2DM remission post-surgery. High values of hsCRP correlate with a low insulin secretory reserve and therefore, a less probability of post-surgical T2DM remission. Adiponectin is an indirect indicator of insulin resistance, a low adiponectin level suggesting a higher degree of insulin resistance. In addition, proinsulin levels, proinsulin/insulin ratio and the visceral fat percentage can be considered valuable markers for predicting the progressive/regressive trend of pathogenic processes linked to T2DM.

## 4. Materials and Methods

We conducted a randomized controlled study type PCCA (collaborative applicable research projects), number PN-II-PT-PCCA-2013-4-2154, on 41 T2DM obese male participants (body mass index, BMI ≥ 30 kg/m^2^), which were randomly selected from a pool of 144 Caucasian patients from all over the country, aged 30–65 years. The study was approved by the local Ethics Committee from ”Ponderas” Hospital and “Prof. N. C. Paulescu” National Institute of T2DM, Nutrition and Metabolic Disease, Bucharest (authorizations for carrying out study activities—aut. No. 412/2.12.2014 and aut. No. 4088/08.12.2014). All patients participating in the study signed an informed consent in accordance with the Helsinki Declaration (1975, revised in 2008); procedures for working with human subjects and biologic samples from them received the favorable opinion of the Ethics Committees of each partner institution. Due to the high cost of LSG surgery in Romania, only a limited number of subjects could be included in our study. Taking this into account, we decided to analyze only male subjects with the same type of obesity (predominantly abdominal obesity), in order to achieve a homogeneous study group. All 41 patients were randomly divided into two groups: (1) patients receiving conventional (non-surgical, antidiabetic) treatment of T2DM and (2) patients undergoing LSG. The randomization of patients included in the study is based on a pseudo-generator created on the Microsoft Visual Studio 2013 Community Edition platform, based on the algorithm described by Donald E. Knuth [45].

The following inclusion criteria were used: sex-male, age 30–65 years, T2DM 1–15 years duration, body mass index (BMI) between 30–50 kg/m^2^ (all subjects had abdominal obesity), possibility of covering the cost of postoperative medication after LSG. The exclusion criteria were: type 1 T2DM, C-peptide < 0.81 ng/mL, HbA1c < 6.5%, anemia (Hb < 10 g/dL), the presence of active liver disease or hepatic dysfunction (hepatitis B or C, cirrhosis), renal disease (serum creatinine > 1.2 mg/dL or glomerular filtration rate GFR < 60 mL/min/1.73 m^2^), malignancies, coronary artery disease with myocardial infarction or stroke in the last 12 months, thyroid and psychiatric pathology, chronic pathology of the digestive tract or the adjacent glands and/or major surgical interventions in the digestive system (gastric/intestinal resections/acute pancreatitis), smoking, alcoholism, drug dependence. No participant had any diagnosed systemic immune disorder and was not known to be taking any form of vitamin supplementation neither at the time of recruitment nor after inclusion in this study or any other treatment with immunosuppressant, corticosteroids and anticoagulant therapy.

Starting from the measured value for resting metabolic rate (RMR) and taking into account the “nutritional pattern” or identified dietary habits, a personalized diet was established for each patient from the conservative treatment group [46]. Specifically, the daily caloric requirement was calculated based on the formula [RMR * 1.3 (sedentary lifestyle) + (10% * RMR) (for the thermal effect of food) −500], thus inducing a 500 kcal daily restriction. In addition, all subjects received lifestyle counseling regarding the increase of physical activity (≥30 min of brisk walking every day or moderate exercise at least 30 min, three to five times per week), limiting alcohol consumption and cessation of smoking, maintaining and monitoring the prescribed oral therapy for T2DM with metformin (1–3 g/daily), hypertension and dyslipidemia according to current guidelines. Unfortunately, with extremely rare exceptions, clinically significant weight loss is generally very modest and transient, particularly in patients with severe obesity [47,48]. The failure rate for these programs was around 95% after one year.

Patients from the surgical group underwent a LSG procedure. Subsequently, they received specific dietary advice (vitamins and minerals supplementation) and down-titration of T2DM medication according to their blood glucose profiles and even removal of antidiabetic oral medication. Patients in the two groups were followed for over one year to observe the impact of LSG in comparison with the group receiving conventional treatment on decreasing body fat mass and main parameters of glucose, lipid and protein metabolism. Based on this multiparametric investigation we were able to particularize the selection algorithm of patients with T2DM and obesity having the highest chances of T2DM remission after correcting obesity taking into account the geographic variation.

Before LSG, at visit V1 (considered the starting point of the study) and then on the occasion of the follow-up visits, i.e., V2 (after six months) and V3 (after 12 months), the below investigations were carried out for all 41 patients.

A complete physical examination was performed and information was gathered about: age, height, weight, abdominal circumference, heart rate, blood pressure, clinical examination of the respiratory, digestive system, clinical information about treatment, duration of T2DM, eating and life style habits.

For all patients, the body composition was determined by the bioimpedance method (using the Body Composition Analyzer Tanita BC-418 MA), thus determining the following parameters: weight, body mass index (kg/m^2^), body fat (%), fat mass (kg), free fat mass (kg), total body water (kg), visceral-fat rating (%).

The resting metabolic rate (RMR) was determined by indirect calorimetry with the COSMED QUARK CPET analyzer (cardio-pulmonary exercise testing).

Blood samples were collected for the following laboratory investigations: baseline biochemical measurements, HbA1c (by high performance liquid chromatography-HPLC method on D10-Bio-Rad Analyzer), blood count (by automatic photometric method on Cell-Dyn 3700 Bio-Rad Analyzer), rapid serologic tests for HIV, hepatitis B and C. glycemia, a lipid (total serum cholesterol, high density lipoprotein HDL-cholesterol, low density lipoprotein LDL-cholesterol, serum triglycerides) and also a hepatic and renal profile (creatinine, urea, uric acid, albumin, total protein, hepatic transaminases and gamma glutamyl transferase GGT) was performed on the Hospitex Diagnostics Eos Bravo Forte biochemistry analyzer using specific reagents according to the manufacturer’s technical datasheets.

Serum insulin, proinsulin, C peptide were determined by enzyme-linked immunosorbent assay (ELISA) using commercially available kits (EIA-2935, EIA-1560, EIA-1293, DRG Instruments, Germany). Absorbance reading (at 450 nm) was performed on ELISA plate reader: MULTISKAN Ex-Thermo Electro Corporation (CV = 2.6%).

Based on blood glucose and insulinemia values, beta cell function was estimated (HOMA% B-homeostasis model assay for β cell function) according to the formula (20 × fasting insulin (µU/mL)/(fasting blood glucose (mmol/L)−3.5%. In addition, insulin resistance was assessed by calculating HOMA-IR (homeostasis model assay for insulin resistance) by the formula: [(fasting blood glucose (mmol/L) × insulinemia (µU/mL)]: 22.5.

Leptin and adiponectin hormones (used to evaluate adipocyte cell function) have also been determined by the enzyme-linked immunosorbent assay (ELISA), using the commercially available kits EIA-2395 and EIA-4177 and the DRG Instruments (Germany), according to the manufacturer’s recommendations. Proinsulin/Adiponectin (P/A) and Proinsulin/Insulin (P/I) ratio was determined by mathematical formula.

Concentrations of hormones involved in the regulation of food intake, including glucagon-like peptide 1 (GLP-1) with anorexic role and ghrelin with orexigenic role, have also been determined by using commercially available ELISA kits (MBS760336 MyBioSource, Inc., BioZyme for GLP-1 kits and EIA-4710 kit for the active (acylated) form, human ghrelin (active) (DRG Instruments, Germany).

Tumor necrosis factor alpha (TNF-α), interleukin 6 (IL-6), high sensitivity C reactive protein (hsCRP) and homocysteine circulating proinflammatory markers were determined by commercially available ELISA kits EIA-4641, EIA-4640, EIA-3954 (DRG Instruments, Marburg, Germany) and MBS260128 MyBioSource Inc (BioZyme).

For all ELISA tests, absorbance reading (at 450 nm) was performed on the automated ELISA plate reader: MULTISKAN Ex-Thermo Electro Corporation (CV = 2.6%).

The oxidative stress profile consisting in the evaluation of “respiratory burst” and the antioxidants enzymes: paraoxonase1 (PON1), superoxide dismutase (SOD), glutathione peroxidase (GPx), catalase (CAT) and also 8-OH-2-deoxiguanosine—marker for the oxidative modification of DNA was investigated at V1 and V2.

Evaluation of oxidant/antioxidant status in the peripheral circulation of patients with obesity and T2DM involved isolation of peripheral blood mononuclear cells (PBMC) and respiratory burst (RB) determination in all patients.

The ability of PBMCs to produce free radicals was determined by measuring the activity of NADPH oxidase Nox2. Mononuclear cells in the peripheral blood were isolated by density gradient centrifugation with Ficoll-PaqueTM Plus (1.0077 g/mL). After centrifugation at 630 g for 30 min the PBMCs were harvested, washed twice and resuspended in 1-mL phosphate buffer saline (PBS). Cell viability was checked with Trypan Blue and was always ≥ 90%. The RB production was monitored by assessing the lucigenin (LG)/luminol (LM) chemiluminescence on PMA stimulation (phorbol-12 myristate-13 acetate)/opsonized zymozan (OZ). Over the isolated PBMCs (0.3 × 10^6^ cells) resuspended in PBS, lucigenin and luminol were added. Spontaneous chemiluminescence monitoring was performed for 15 min, after which RB was initiated by the addition of PMA/OZ; the chemiluminescence peak was recorded with the Luminoskan Ascent Thermo Scientific luminometer. The results were expressed as relative chemiluminescence units (RLU).

Determination of erythrocyte glutathione was performed on total blood samples (200 μL) after precipitation with metaphosphoric acid solution, disodium salt dihydrate (Na_2_EDTA) and sodium chloride (NaCl). After centrifugation at 4000 rpm, over 250 μL of the supernatant was added 1 mL of 0.3-M phosphate buffer and 125 μL DTNB (Ellman reagent: 5,5′-dithiobis-2-nitrobenzoic acid) over 15 min at 4 °C. Absorbance was read at 405 nm and the results were expressed as μg/g Hb.

Antioxidant enzymes PON1, arylesterase (PON1phe) and lactonase 1 (PON1dhe) activities were measured toward 1-mM phenylacetate in 20-mM Tris/HCl pH 8.0 or 1-mmol/L DHC, respectively. The reaction was started by the addition of the serum and the increase in absorbance was read at 270 nm using a UV-Vis spectrophotometer. Blanks were included to correct the spontaneous hydrolysis of substrate. One unit (U) of arylesterase (PON1phe) is defined as 1 pmol of p-nitrophenol hydrolyzed per minute using the extinction coefficient of 1310 M-1 cm-1 while one unit of lactonase activity (PON1dhe) is equal to 1 µmol of DHC hydrolyzed/mL/min using the extinction coefficient of 1295 M^−1^ cm^−1^. The intra- and inter-assay coefficients of variation were < 5% in all tests. SOD and GPx were measured by using kit no. K9120 for SOD and kit no. 30-7031 for GPx, respectively (Sigma-Aldrich Co. LLC., St Louis, USA), according to the producer recommendations. Results are expressed as U/g Hb. The method of determining the activity of erythrocyte CAT is based on the decrease in the absorbance of H_2_O_2_ at 240-nm with decreasing concentration. The initial hemolysate obtained was diluted with 50-mM phosphate buffer (pH 7.0) and the reaction was initiated with H_2_O_2_. The results were expressed in k/g Hb.

Determination of the serum 8-OH-2-deoxyguanosine (8-OH-2dG) concentrations was performed with a competitive ELISA kit (Abcam) using a monoclonal antibody specific for 8-OH-2dG and an acetylcholinesterase (AChE) conjugated 8-OH-2-dG tracer according to the manufacturer instructions; the substrate for AChE is contained in the developing reagent, and the yellow enzymatic reaction product was determined spectrophotometrically at 412 nm.

To avoid possible postoperative complications, in addition to the conventional treatment group, patients randomized to the surgical treatment group have undergone preoperative investigations: barium transit, upper digestive endoscopy under intravenous anesthesia, cardiac evaluation, spirometry, abdominal ultrasound scan, chest X-ray.

Out of the 21 patients in the conventional group and 20 patients in the surgical group initially included in the study, 19/17 reached V2 and 19/15 patients were present at the last visit (therefore five patients decided to withdraw from study).

Statistical Analysis

All statistical analyses were conducted using the SPSS version 20.0 software. We used mean ± standard deviation (SD)/standard error of mean (SEM) to describe continuous variables with a normal distribution and median with interquartile range (in brackets) for variables with skewed distribution. Some variables were converted logarithmically to normalize their distribution before analysis. Results for variables that did not follow a normal distribution were presented as median and interquartile range (IQR) (25–75). Paired Student’s *t*-tests and Wilcoxon signed rank test were used to compare data from the two treatment groups (CTG versus LSG), while the 2-way ANOVA with Bonferroni post hoc test was used when analyzing the visits main effect (V1 versus V2 versus V3). A value of *p* < 0.05 and *p* < 0.001 was considered statistically significant.

## 5. Conclusions

This prospective, multiparametric study confirms the complex picture of T2DM and the close relationship between T2DM, obesity, oxidative stress, inflammation—without being able to specify the time sequence in which they occur in diabesity. LSG appears to be a promising alternative for diabesity therapy, even in non-morbidly obese patients. Given the significant spatial heterogeneity in the etiopathogenic profile and prevalence of T2DM and obesity, geographically based data are fundamental for implementing the most appropriate algorithms for the selection of patients underlying LSG. The results obtained in patients after surgery compared favorably to those in patients with standard medical therapy.

The limitations of this study are the small number of patients enrolled in the study and the short duration of the monitoring period. However, these limitations are offset by the large panel of biomarkers evaluated over one year that could offer an overview of the metabolic, inflammatory and oxidative stress profile of T2DM obese patients undergoing different therapeutic interventions. Additionally, further long-term studies are needed on a larger population of T2DM associated with obesity cases.

## Figures and Tables

**Table 1 metabolites-10-00308-t001:** Comparison of the hormonal levels determined at V1, V2 and V3 for the conventional treatment (CTG) versus LSG group; 2 way-ANOVA with Bonferroni post hoc (data shown as median with IQR, *p*-value shows interaction effect, significant results are marked with *, ** or *** and show the significant differences observed at V2 and V3 compared to V1, if not marked, results are not statistically significant).

Hormones	CTG	LSG	CTG	LSG	CTG	LSG	*p*-Value
	V1	V1	V2	V2	V3	V3	
	*n* = 21	*n* = 20	*n* = 17	*n* = 19	*n* = 15	*n* = 19	
**Insulin (μUI/mL) ***	22.12 (23.41)	18.92 (24.35)	17.43 (12.71)	6.67 (5.68)	14.13 (31.83)	5.08 (4.17)	0.5055
**HOMA-IR (Homeostatic Model Assessment for Insulin Resistance) ***	8.12 (14.02)	8.44 (10.56)	7.35 (5.70)	1.68 (1.70)	6.32 (11.93)	1.19 (0.80)	0.1128
**HOMA (Homeostatic Model Assessment)-β ***	90.68 (52.24)	60.16 (84.72)	86.76 (131.14)	72.03 (97.63)	67.54 (146.79)	61.25 (49.27)	0.7021
**Proinsulin (pmol/L) ***	6.42 (19.76)	5.86 (6.59)	4.84 (6.60)	0.87 (0.71)	3.06 (5.31)	0.70 (0.87)	0.5946
**Proinsulin/Insulin ***	0.44 (0.59)	0.32 (0.29)	0.21 (0.73)	0.12 (0.12)	0.19 (0.15)	0.12 (0.18)	0.9399
**C Peptide (ng/mL) ***	9.82 (2.53)	9.72 (1.33)	7.66 (2.10)	7.39 (2.09)	14.13 (31.83)	4.19 (1.70)	**0.0251** **** V3**
**Leptin (ng/mL) ***	15.50 (7.66)	8.45 (6.39)	10.16 (12.78)	2.62 (1.29)	8.62 (7.45)	2.65 (0.79)	***p* < 0.0001** ***** V2, ** V3**
**Adiponectin (μg/mL) ***	4.82 (9.28)	2.09 (1.09)	3.98 (6.50)	3.63 (5.58)	3.52 (2.61)	7.53 (7.42)	0.0082 *** V3**
**GLP (Glucagon-like peptide)-1 (ng/mL) ***	47.13 (4.85)	47.40 (12.17)	50.54 (9.11)	16.12 (17.61)	–	–	*p* < 0.0001 ***** V2**
**Ghrelin (pg/mL) ***	100.47 (47.2)	117.40 (42.9)	136.75 (62.89)	94.42 (15.72)	140.48 (115.6)	84.82 (32.86)	**0.0017** ***** V3**

**Table 2 metabolites-10-00308-t002:** Comparison of proinflammatory markers levels after 6 and 12 months of treatment for both groups.

Proinflammatory Markers	CTG (Control Group)		*p*	LSG (Laparoscopic Gastric Sleeve Group)		*p*
	V2	V3		V2	V3	
	*n* = 21	*n* = 15		*n* = 20	*n* = 19	
**IL-6 ***	**5.79 (13.41)**	11.51 (9.04)	**0.08325**	3.34 (6.26)	11.51 (11.47)	**0.0004**
**TNFα ***	0.73 (1.12)	3.27 (3.56)	**0.00116**	0.65 (0.90)	2.90 (3.33)	**0.003342**
**Homocysteine ***	1.94 (0.95)	1.94 (1.53)	**0.0637 ^1^**	1.96 (0.88)	2.36 (0.71)	**0.1447 ^1^**
**hsCRP ***	6.34 (6.21)	12.33 (9.35)	**0.0946 ^1^**	9.47 (4.85)	1.31 (2.48)	**0.0006447 ^1^**

* median and IQR (in brackets); ^1^ Wilcoxon signed rank test.

**Table 3 metabolites-10-00308-t003:** Oxidative stress markers in diabetic patients at baseline (V1) and after 6 months (V2).

Oxidative Stress Markers	Baseline			6 Months	*p*-Value
	CTG	LSG	CTG	LSG	
	*n* = 21	*n* = 20	*n* = 17	*n* = 19	
**RB LM/PMA (maximum RLU)**	0.06 ± 0.01	0.19 ± 0.05	0.09 ± 0.02	0.17 ± 0.04	**ns**
**RB LG/PMA (maximum RLU)**	0.01 ± 0.00	0.01 ± 0.00	0.01 ± 0.00	0.01 ± 0.00	**ns**
**RB LM /OZ (maximum RLU)**	0.12 ± 0.03	0.33 ± 0.09	0.23 ± 0.06 *	0.37 ± 0.00	**<0.05**
**RB LG/OZ (maximum RLU)**	0.01 ± 0.00	0.02 ± 0.00	0.02 ± 0.00	0.02 ± 0.00	**ns**
**GSH (μg/g Hb)**	0.27 ± 0.01	0.26 ± 0.01	0.28 ± 0.01	0.23 ± 0.00 **	**<0.05**
**8-OH-2dG (ng/mL)**	10.31 ± 0.54	13.10 ± 2.14	10.15 ± 1.82	12.04 ± 0.70	**ns**

RB—respiratory burst; PMA—phorbol 12-myristate 13-acetate; ZO—opsonized zymosan; LM—luminol; LG—lucigenin; GSH—glutathione—8-OH-2dG—8-hydroxy-2-deoxyguanosine; maximum—the maximal peak value; RLU—relative chemiluminescence units. Data expressed as mean ± SEM (standard error of the mean). The *p*-value refers to the comparison between CTG and LSG after 6 months; * and ** denote *p* < 0.05 and *p* < 0.001 when comparing 6 months to baseline within each group; ns = not significant.

**Table 4 metabolites-10-00308-t004:** Levels of antioxidants markers in erythrocytes obtained from the patients included in the study.

Antioxidant Markers	V1		V2		V3	
	CTG	LSG	CTG	LSG	CTG	LSG
	(*n* = 21)	(*n* = 20)	(*n* = 17)	(*n* = 19)	(*n* = 15)	(*n* = 19)
**GPx (U/g Hb)**	8.33 ± 0.49	6.26 ± 0.5	14.18 ± 0.96 **	8.80 ± 0.1*	6.16 ± 0.6 *^,$$^	8.32 ± 0.12 ^$^
**SOD (U/g Hb)**	605.02 ± 14.6	626.47 ± 14	665.58 ± 35.9	586.16 ± 40.5	465.52 ± 38.7 *^,$$^	515.41 ± 28.05
**CAT (k/g Hb)**	9.28 ± 0.46	9.96 ± 0.56	14.77 ± 1.8 *	15.50 ± 1.48 *	13.80 ± 0.8*	19.95 ± 1.22 ^$^
**PON1dh (μmol/mL/min)**	2.15 ± 0.04	2.34 ± 0.06	2.26 ± 0.09	2.31 ± 0.05	2.04 ± 0.06	2.45 ± 0.08
**PON1phe (μmol/mL/min)**	73.54 ± 2.5	73.94 ± 2.8	75.94 ± 1.7	73.94 ± 2.8	74.34 ± 2.2	71.92 ± 1.88

GPx—glutathione peroxidase, SOD—superoxide dismutase, CAT—catalase, PON1dh—paraoxonase1, * and ** represent *p* < 0.05 and *p* < 0.001, respectively, initially comparing 6 or 12 months for each group; ^$^ and ^$$^ represents *p* < 0.05 and *p* < 0.001 comparing 6 and 12 months between groups (CTG and LSG).

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
