# Peer review of "Evolution of Inflammatory and Oxidative Stress Markers in Romanian Obese Male Patients with Type 2 Diabetes Mellitus after Laparoscopic Sleeve Gastrectomy: One Year Follow-Up"

_metabolites, 2020, doi:10.3390/metabo10080308_

Round 1

Reviewer 1 Report

The authors investigated if LSG is superior in T2DM remission compared to conventional (non-surgical antidiabetic) treatment in Romanian obese patients. Their data demonstrated a significant superior effect of LSG at 6- and 12-months post-intervention. In general, the data support the currently available scientific data. However, it is done in a specific ethnic population, which is always a valuable contribution to the scientific knowledge. However, there are significant limitations in data analyses and presentation, which diminish the value of this work. Please see below for specific point-by-point comments; however, please be advised that much more needs to be done to improve the manuscript.

Major comments:

  1. In general, there is lots of irrelevant information such as the factors that contribute to increased development of diabetes in Romania (lane 64-65). The detailed description of LSG is irrelevant (Lane 71-77:). Discussion also has lots of information on parameters that are irrelevant to the data presented in this work. I would suggest editing by removing the irrelevant information. In addition, some language and scientific editing is required throughout the manuscript. There are several paragraphs that consist of 1 sentence only, which is grammatically incorrect.
  2. The 4th paragraph of introduction (lanes 78-85) is completely irrelevant, and thus can be omitted. In my opinion, if the comments in #1 are addressed, the paragraphs 2-5 can be compressed into one paragraph.
  3. Lanes 96-99: were there any statistically significant differences in the age and duration of the T2DM between the two groups. P-values need to be added.
  4. Statistical analyses to assess the changes in repeated measures should have been done using Two-way ANOVA repeated measures? This is a major limitation of data analyses and presentation. If the stats are done using Two-way ANOVA repeated measures, the Tables 1-3 can be combined by omitting the duplicative presentation of V1 data.
  5. Lanes 157-161: It is mentioned that there were significant changes in inflammatory markers at 6-month time point (V2); however, no actual data is being presented (e.g., Table 6).
  6. Lane196-197: The authors state that diabesity is positively correlated with the intensity of oxidative stress and pro-inflammatory status. However, no data on correlation analyses are being presented.
  7. Lane 204: It is stated that “The antioxidant enzymes GPx and CAT increased after 6 months in both groups”; however, I was not able to locate these data. Although, the method section contains the description of approach used to measure these enzymes. Table 6, GSH is presented, which seems to have decreased in LSG group at 6 months; however, no discussion about these results is being presented. Lane 206: SOD and PON1 are discussed; however, no data can be located.

Minor comments:

  1. Lane 47: What kind of “C Peptide parameters” specifically?
  2. Lane 48 – What do you mean by “monitoring the disease”?
  3. Lane 53: remove “mellitus (T2DM)”. Moreover, if the manuscript is about T2DM, please use this abbreviation throughout instead of “diabetes” (e.g., lane 56, 57, 58, etc). Or introduce that both “T2DM” and “diabetes” will be used interchangebly.
  4. Lane 66-70: If there is no difference in the risk of development of T2DM depending on the obesity class I would suggest editing “20% of Romanian people with class 1 and 2 obesity or with morbid obesity (class 3) develop T2DM” by removing the description of classes. Having the following sentence about the morbid obesity is acceptable with edits that morbid obesity worsens the situation.
  5. Table 1: “Nutritional” – is this needed?
  6. Lane 190: edit “a decrease of pro-inflammatory and oxidative stress markers” to “decreases of pro-inflammatory and oxidative stress markers”.

Reviewer 2 Report

This study Picu et al. evaluated the impact of LSG on male patients with T2M and obesity. They have used multiparametric parameters including metabolic rate, general biochemical status, adipocytes profile, gastrointestinal hormones levels, pro-inflammatory, oxidant, and antioxidant profile, etc. before, as well as 6 months and 12 months after the LSG intervention. Simultaneously, they also used conventional (non-surgical, antidiabetic) approach in one of their patients with T2M and obesity. The authors want to emphasize if LSG treatment is better than CTG treatment and compared the parameters after CTG and LSG intervention. In general, this is a very interesting and well-designed study and we need more study like this to evaluate the impact of multiple interventions. I also understand that this is a pilot-based study and there are many limitations to get data from the human-based study. However, the following comments/questions need to be answered before this can be considered for the publication.

  1. Most of the parameters including waist circumference and BMI are lowered with both treatments (extent is different) however, some parameters are specific to one vs another treatment. The authors need to provide a general conclusion about how one treatment is better than another considering all the parameters into consideration?
  2. Total cholesterol is significantly lowered with CTG but not in LSG group after the treatments. However, TG is lowered and HDL increased in LSG group after the treatment. How authors are going to explain these results?
  3. In general, with few exceptions, LSG group showed more improvement of diabetic linked parameters toward normal including physiological, metabolic, inflammation, oxidative stress, and hormonal labels. However, mechanistic detail is lacking how a single parameter or multiple parameters are working in one treatment vs another in mitigating diabetic risks.
  4. These improved parameters with both treatments never compared with non-disease control. How is the improvement in each cardiac parameter compared to age-matched control? The authors did not enroll in the control, they can look at data of available control group.
  5. There are many risks associated with LSG treatments however, these risks are mainly ignored and need to include in the discussion.
  6. It is unclear the treatments are for obesity, diabetes, obesity-linked diabetes, or everything. There are many differences in obesity and diabetes. The authors need to did not try to rule out that difference and missing two diseases together. In other words, what would have been authors' expectations in they are using these interventions either on diabetic or obese patients?
  7. The treatment in the non-surgical group need more clarification in the method section
  8. Although, the sample size is very small the authors noticed more risks in older patients, and what were the impacts of both treatments in the old vs young patients?
  9. the entire manuscript needs to be carefully checked for the typo and grammar issues.

Round 2

Reviewer 1 Report

Thank you for addressing the comments, I believe that the manuscript has significantly improved. Please address a few minor comments mentioned below.

Minor comments:

  1. Lane 191: Edit the following sentence: “levels in LSG patients compared with the conventional treatment group [23].
  2. Grammatically there cannot be one-sentence paragraphs (Lane 316-317).
  3. I would suggest editing the discussion section on oxidative stress (lanes 275-317). The discussion is very scattered.
  4. The common formats to report the p-values are “p<0.05” and “p<0.001”. I would suggest using these formats instead of “0.0001428” or “<0.000001”.
  5. Supplemental Table 1: Introduce “CTG” abbreviation in the footnotes of the table, and remove “Conventional Treatment Group” from the top row of the table.
  6. Please recheck the manuscripts for grammatical errors.

Author Response

  1. Lane 191: Edit the following sentence: “levels in LSG patients compared with the conventional treatment group [23].

The sentence was edited as suggested.

  1. Grammatically there cannot be one-sentence paragraphs (Lane 316-317).

             This was modified as suggested.

  1. I would suggest editing the discussion section on oxidative stress (lanes 275-317). The discussion is very scattered.

             This section was improved as suggested.

  1. The common formats to report the p-values are “p<0.05” and “p<0.001”. I would suggest using these formats instead of “0.0001428” or “<0.000001”.

             The p-values were reported as suggested (“p<0.05” and “p<0.001”).

  1. Supplemental Table 1: Introduce “CTG” abbreviation in the footnotes of the table, and remove “Conventional Treatment Group” from the top row of the table.

             The CTG abbreviation was introduced instead of the text in Table 1.

  1. Please recheck the manuscripts for grammatical errors.

             The manuscript was checked for grammar errors and corrected where necessary.